# Metformin exposure, maternal PCOS status and fetal venous liver circulation: A randomized, placebo-controlled study

Sindre Grindheim[1]*, Cathrine Ebbing[1,2], Henriette Odland Karlsen[1], Svein Magne Skulstad[3], Francisco Gómez Real[1,2], Marianne Lønnebotn[3], Tone Løvvik[4,5], Eszter Vanky[4,5‡], Jørg Kessler[1,2‡]

1 Department of Obstetrics and Gynaecology, Haukeland University Hospital, Bergen, Norway, 2 Department of Clinical Science, University of Bergen, Bergen, Norway, 3 Department of Global Public Health and Primary Care, University of Bergen, Bergen, Norway, 4 Department of Clinical and Molecular Medicine, Faculty of Medicine and Health Sciences, Norwegian University of Science and Technology, Trondheim, Norway, 5 Department of Obstetrics and Gynaecology, St. Olav's Hospital, Trondheim University Hospital, Trondheim, Norway

‡ EV and JK shared last authorship on this work.
* sindre.grindheim@helse-bergen.no

**Data Availability Statement:** All relevant data are within the manuscript and its Supporting Information files.

## Abstract

### Background

Metformin is prescribed to women with polycystic ovary syndrome (PCOS) to prevent pregnancy complications. Children exposed to metformin vs. placebo in utero, have increased head circumference at birth and are more overweight and obese at 8 years of age. Also, maternal PCOS-status seems to alter the long-term cardio-metabolic health of offspring. We hypothesized that the long-term effects of metformin-exposure and/or maternal PCOS may be mediated by circulatory adaptations during fetal life.

### Material and methods

This is a sub-study of a larger double-blinded, placebo-controlled trial, where women with PCOS were randomized to metformin (2g/day) or placebo in pregnancy, a total of 487 women. A sub-group of participants (N = 58) took part in this sub-study and had an extended ultrasound examination at gestational week 32, including blood flow velocity and diameter measurements of the umbilical vein (UV), the ductus venosus (DV) and the portal vein (PV). Blood flow volume was calculated and adjusted for estimated fetal weight (EFW) (normalized flow). Metformin exposed fetuses were compared to placebo exposed fetuses. Fetuses of mothers with PCOS (metformin [n = 30] and placebo [n = 28]) were compared to a low-risk reference population (N = 160) by z-score statistics.

### Results

There was no difference in fetal liver flow between metformin vs. placebo-exposed fetuses. Fetuses of mothers with PCOS had higher EFW (0.63 [95% CI 0.44–0.83] p<0.001), lower

**Funding:** 1. EV recieved grant form the The research Council of Norway for the project. 2. Project title: Metformin treatment of pregnant PCOS women and prevention of preterm birth. Projectnumber.: 213497/H10 3. https://www.forskningsradet.no/en/ 4. The research Council of Norway 5. The funders had no role in study design, data collection and analysis, decision to publish, or preparation of the manuscript.

**Competing interests:** The authors have declared that no competing interests exist.

normalized UV, DV, PV, and lower total venous liver blood flows than the reference population.

## Conclusion

Metformin during pregnancy did not affect fetal liver blood-flow. In our population, maternal PCOS-status was associated with reduced total venous liver blood-flow, which may explain altered growth and metabolism later in life.

## Introduction

In unselected populations, nearly 17% of women have polycystic ovary syndrome (PCOS) [1]. Women with PCOS have increased risk of adverse pregnancy outcomes such as gestational diabetes (GDM), preterm delivery, preeclampsia and small for gestational age new-borns [2]. Metformin, an anti-diabetic drug, is increasingly used to treat both GDM and PCOS. Metformin reduces hepatic glucose production, insulin resistance, and inhibits lipogenesis in the non-pregnant state [3]. In addition to the liver, metformin affects a variety of organs, such as pancreas, gonads and intestinal microbiota [4, 5]. It also prevents late miscarriage and preterm deliveries in mothers with PCOS [6].

Metformin crosses the placenta and has been detected in therapeutic concentrations in umbilical cord blood [7]. Metformin has the potential to affect embryonic cells and fetal tissues [8], and has an anti-folate effect in supra-therapeutic concentrations [9]. These findings have raised concern that the drug might inhibit cell growth and result in a state of nutrient restriction in the fetus, causing metabolic diseases later in life by epigenetic programming [10]. Follow-up studies of children born to women with PCOS and exposed to metformin in utero demonstrated increased BMI from 6 months to 8 years of age, more central fat distribution and more obesity [11, 12]. Maternal PCOS status also seems to affect the long-term health of the offspring [11, 13]. There is, however, little knowledge about the mechanisms of in-utero metformin-exposure to explain possible long-term effects on exposed offspring. Receiving 80% of the umbilical vein flow, the liver plays a major role in the regulation of fetal growth [14, 15]. Fetal venous liver flow seems to be an adaptation to present and future metabolic challenges. It is regulated by several external factors, such as maternal diet [16], body composition [16], weight gain [17] and glycaemic control during pregnancy [18]. We hypothesized that in women with PCOS, intrauterine exposure to metformin may reduce total liver blood flow.

The aim of the present study was to compare fetal venous liver blood flow in both metformin vs. placebo-exposed fetuses of mothers with PCOS, and all fetuses of PCOS mothers to a low-risk reference population.

## Material and methods

The PregMet 2 study, a Nordic multi-center RCT, aimed to explore whether metformin reduced late miscarriages and preterm births in women with PCOS. Participants (N = 487) were enrolled to the study from October 2012 to December 2016, and recruitment were based on diagnosis fulfilling the Rotterdam criteria [19]. Participants were recruited in the first trimester of pregnancy, gave informed written consent, and were randomized to metformin 2g/placebo during pregnancy. Detailed information on recruitment and randomization is described elsewhere [6]. Identical routines for recruitment and randomization were applied in

the sub-study presented here. Participants on pre-conception and early pregnancy metformin treatment had a wash-out period of at least 7 days before randomization to metformin or placebo. Study medication was given as metformin 500 mg x 2 daily or placebo during the first week and increased to 1000 mg x 2 daily or placebo, from week two until delivery. Treatment was started, in the first trimester, as soon as possible after the inclusion visit, but latest seven days after the inclusion visit. The majority of the participants started their study medication the day following inclusion, and 56/61 patients had an adherence of more than 70%.

## Sub-study and reference populations

From October 2013 to December 2016, women included in the PregMet2 study at Haukeland University hospital were invited to participate in a sub-study with additional ultrasound examinations (the CircMet study). Of the 80 women included to the PregMet2 study in this period in Bergen, 61 (76%) gave separate written consent and were scheduled for an extended ultrasound examination at gestational week 32 (Fig 1).

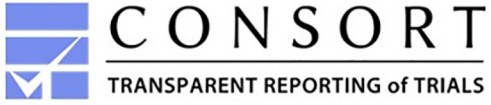

**CONSORT 2010 Flow Diagram**

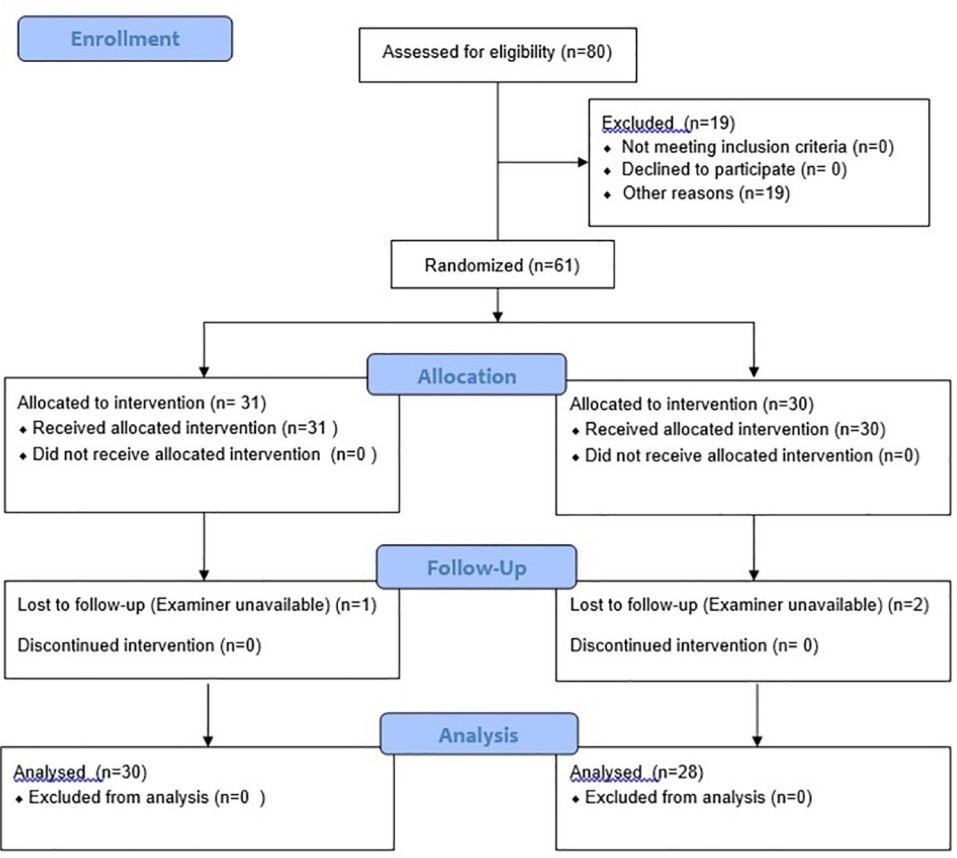

**Fig 1. Flow chart of inclusion, randomization and dropout.**

The reference population for the Doppler and flow parameters used in the present study consisted of healthy pregnant women without previous pregnancy complications and a normal result of the 2nd trimester routine ultrasound scan. Details on the reference population are described elsewhere [17, 20, 21], and shown in Tables 1 and 2.

## Ultrasound examinations

Due to limited resources, we performed a cross-sectional ultrasound study. We chose to perform the measurements at gestational week 32 as we expected that the chance to detect circulatory maladaptation would be higher in late pregnancy. The fetal vessel diameter increases as the fetus grows, making measurements easier. Women with PCOS often have abdominal adiposity, making ultrasound measurements of small structures challenging earlier on. We did not choose gestational week 36, as the frequency and duration of fetal breathing movements increase towards term, interfering with the standardized way to perform Doppler ultrasound measurements of flow velocity. An ultrasound examination of 60 min duration at 32 weeks is less probable to be interrupted due to maternal discomfort in the supine position at 32 than 36 weeks.

Voluson E6 and E8 machines (2–5 MHz curved linear abdominal transducer) were used for the study. No examination lasted more than 60 minutes, and included measurements of fetal biometry (head circumference (HC), abdominal circumference (AC) and femur length (FL)), blood flow velocity and pulsatility index of the uterine arteries (UtA), the umbilical artery (free loop) (UA), the middle cerebral artery (MCA), the hepatic artery (HA). The vessel diameter (D) and blood flow velocities were measured at the intraabdominal portion of the umbilical vein (UV), the inlet of the ductus venosus (DV) and the main portal vein (PV). Further, the blood flow velocity in the left portal vein was measured. The techniques of measurement and calculations are described in details elsewhere [17] and were identical to those used to establish the reference ranges in the low-risk reference population.

Volume blood flow (Q (ml/min)) in the different vessels (Fig 2) were calculated using the measured vessel diameter (D), time averaged maximum velocity (TAMXV) and velocity profile constant for each vessel (h) as $Q = \pi(D/2)^2 h \text{TAMXV}$ ($h$: velocity profile. h = 0.5 for UV [22, 23] and PV [20]; h = 0.7 for DV [24, 25]). The total venous blood supply of the liver ($Q_{liver}$) was calculated as: $Q_{liver} = (Q_{UV} - Q_{DV}) + Q_{PV}$. The term normalized flow refers to an adjustment of flow volume for estimated fetal weight (flow/kg EFW) [26]. The ultrasound examiner in the CircMet study was blinded for the treatment allocation.

## Statistical analyses

Since this study was a sub-study of a larger RCT, we compared baseline characteristics between the groups to detect differences with biological potential to affect outcomes. Data were analysed by t-test for continuous variables, and chi square test for the categorical variables. Z-score statistics and independent sample t-test with 2-tailed p-values were used to compare Metformin exposed fetuses [N = 30] to placebo exposed fetuses [N = 28]. Secondly, all fetuses born to the PCOS mothers (metformin or placebo treated) were compared to the reference population [17, 21]. Differences in baseline between the groups, which potentially could account for flow differences (BMI and maternal blood pressure), were adjusted for by a General Linear Model analysis for each z-score using randomization as fixed factor and the maternal parameters as covariates. Multicollinearity was assessed and accounted for. The level of significance was defined as $p < 0.05$. The statistical analyses were performed using the IBM Corp. Released 2016. IBM SPSS Statistics for Windows, Version 24.0. Armonk, NY: IBM Corp.

**Table 1. Maternal baseline characteristics and outcomes for women with PCOS randomized to metformin or placebo, and the reference population for the Doppler and flow variables.**

| At baseline | PCOS-met N = 30 | PCOS-plac N = 28 | PCOS-tot N = 58 | Reference N = 160 | p-value metformin vs. placebo | p-value placebo vs. reference | p-value PCOS tot vs reference |
|---|---|---|---|---|---|---|---|
| Age (years) | 29 ±4 | 30 ±4 | 30 ±4 | 29 ±4 | 0.564 | 0.288 | 0.180 |
| Weight (kg) | 78 ±14 | 72 ±14 | 75 ±14 | 66 ±12 | 0.109 | **0.026** | **<0.001** |
| BMI (kg/m$^2$) | 27.7 ±5.5 | 25.3 ±5.2 | 26.6 ±5.4 | 23.4 ±3.8 | 0.091 | **0.018** | **<0.001** |
| Systolic BP (mmHg) | 109 ±8 | 99 ±12 | 104 ±11 | n/a | **0.001** | | |
| Diastolic BP (mmHg) | 67 ±7 | 60 ±7 | 63 ±7 | n/a | **0.007** | | |
| Metformin at conception (%) | 12 (40.0) | 9 (32.1) | 21 (36.2) | 0 | 0.534 | | |
| PCOS phenotype | | | | | 0.231 | | |
| A (OA, HA, PCO) | 19 | 21 | 40 (69.0) | | | | |
| B (OA, HA) | 0 | 1 | 1 (1.7) | | | | |
| C (OA, PCO) | 7 | 6 | 14 (22.4) | | | | |
| D (HA, PCO) | 4 | 0 | 4 (6.9) | | | | |
| Mode of conception n (%) | | | | | | | |
| Spontaneous | 23 (76.6) | 18 (64.3) | 41 (70.7) | 158 (98) | 0.301 | **<0.001** | **<0.001** |
| Clomiphene citrate | 10 (33.3) | 9 (32.1) | 19 (32.8) | 1 (1) | 0.923 | **<0.001** | **<0.001** |
| IVF/ICSI | 1 (3.3) | 4 (14.3) | 5 (8.6) | 1 (1) | 0.138 | **<0.001** | **0.006** |
| Others | 3 (10.0) | 0 (0) | 3 (5.2) | | 0.086 | | |
| Former miscarriage n (%) | 13 (43.3) | 8 (28.6) | 21 (36.2) | n/a | 0.242 | | |
| Smoking n (%) | 0 | 0 | 0 | 16 (10) | | **0.043** | **0.036** |
| Caucasian n (%) | 30 (100.0) | 24 (85.7) | 54 (93.1) | 160 (100) | 0.203 | 0.237 | **0.014** |
| Gestational age at inclusion (days) | 80 ±11 | 76 ±10 | 78 ±10 | n/a | 0.242 | | |
| Parity n (%) | | | | | 0.482 | 0.169 | 0.076 |
| 0 | 14 (46.7) | 11 (39.3) | 25 (43.1) | 93 (58.1) | | | |
| 1 | 13 (43.3) | 11 (39.3) | 24 (41.4) | 42 (26.2) | | | |
| 2+ | 3 (10.0) | 6 (21.4) | 9 (15.5) | 25 (15.6) | | | |
| **Maternal outcomes** | | | | | | | |
| Gestational weight gain [†] (kg/week) | 9.5 ±4.8 | 11.8 ±4.9 | 10.6 ±5.0 | 14.5 ±4.9 | 0.071 | **0.006** | **<0.001** |
| F-glucose gw 28 (mmol/l) [‡] | 4.5 ±0.5 | 4.5 ±0.4 | 4.5 ±0.4 | n/a | 0.870 | | |
| 2h-glucose gw 28 (mmol/l)[*] | 6.9 ±1.4 | 6.3 ±1.6 | 6.6 ±1.5 | n/a | 0.169 | | |
| GDM[¶] n (%) | 12 (40.0) | 6 (21.4) | 18 (31.0) | 0 | 0.149 | | |
| GA at GDM[¶] (weeks) | 26 ±7 | 23 ±9 | | | 0.336 | | |

Independent sample t-test, Chi square test

Data are given as mean +/- SD, or as absolute number (percent).

gw–Gestastional week, GDM–gestational diabetes mellitus, GA–gestational age, OA–oligo-/amenorrhoea, HA–clinical and/or biochemical hyperandrogenism, PCO–polycystic ovaries on ultrasound

**Significant values in bold**

[†]Total weight gain from inclusion to last known weight before delivery

[‡]Missing data from one patient in both groups

[*]Missing data from three patients in the metformin and five in the placebo-group

[¶]Gestational diabetes was an exclusion criterion for the reference population.

**Table 2. Fetal characteristics and neonatal outcomes.**

| | PCOS-met N = 30 | PCOS-plac N = 28 | PCOS-tot N = 58 | Reference N = 160 | P-value Metformin vs placebo | P-value Placebo vs reference | P-value PCOS tot vs reference |
|---|---|---|---|---|---|---|---|
| GA at delivery (days) | 277 ±13 | 281 ±9 | 279 ±12 | 282 ±10 | 0.248 | 0.318 | 0.025 |
| Birth weight (grams) | 3546 ±579 | 3648 ±483 | 3606 ±523 | 3554 ±481 | 0.471 | 0.345 | 0.479 |
| Birth weight (z-score) | 0.28 ±0.88 | 0.24 ±0.90 | 0.26 ±0.88 | | 0.869 | | |
| HC at delivery (cm) | 35.4 ±1.8 | 35.5 ±1.3 | 35.5 ±1.6 | | 0.714 | | |
| Placenta weight (grams)¶ | 689 ±124 | 709 ±173 | 700 ±146 | 657 ±131 | 0.615 | 0.071 | 0.041 |
| Apgar score <7 at 5 min† n (%) | 1 (4) | 1 (4) | 2 (3) | 0 | 1.000 | 0.056 | 0.019 |
| Transfer to NICU‡ n (%) | 3 (10.0) | 1 (3.6) | 4 (6.9) | 7 (4.4) | 0.612 | 0.404 | 0.505 |
| GA at examination (days) | 227 ±5 | 227 ±5 | 227 ±5 | | 0.917 | | |
| Estimated fetal weight at examination (gram) | 2285 ±244 | 2188 ±169 | 2236 ±213 | | 0.080 | | |
| Estimated fetal weigh at examination (z-score) | 0.78 ±0.71 | 0.48 ±0.72 | 0.63 ±0.72 | | 0.122 | | |

Independent sample t-test, Chi square test.

Date are given as mean +/- SD, or as absolute number (percent).

Significant values in bold

GA–gestational age, HC: Head circumference, NICU–neonatal intensive care unit

¶ Missing data from one patient in the metformin group

† Missing data from two patients in the metformin group

‡ Neonatal intensive care unit.

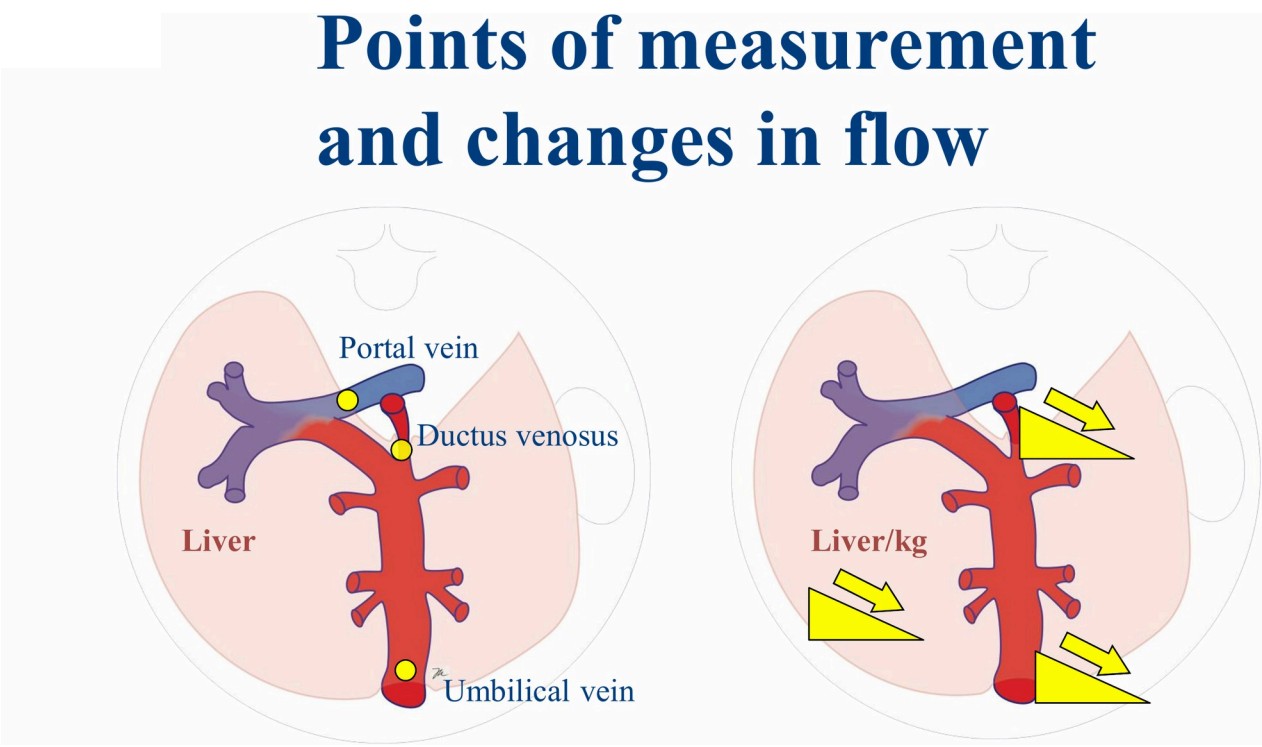

**Fig 2. Illustration of fetal vessels and their changes in blood flow.**

## Ethics

The Regional Committee for Health Research Ethics of Central Norway approved both the PregMet2 study and the CircMet, sub-study (REK 2011/1434) [6]. Written informed consent was obtained from each participant before inclusion both to the PregMet2 and to the present CircMet sub-study. The declaration of Helsinki was followed throughout the study.

## Results

The maternal and neonatal characteristics of the PCOS and reference populations are given in Tables 1 and 2. At baseline, despite randomization, women with PCOS randomized to metformin (PCOS-met) had higher systolic and diastolic blood pressure, and had a borderline significant higher BMI, compared with those in the placebo group (PCOS-plac). These baseline differences between the groups in the CircMet sub-study were not evident in the main study (PregMet2). Metformin vs. placebo treatment resulted in less maternal weight gain in both the main study and in the CircMet sub-study. Baseline characteristics and outcomes were otherwise comparable, including the prevalence of different PCOS phenotypes [27].

Compared to the reference population, women with PCOS had more gestational diabetes, higher BMI and smoked less (Table 2).

Adjusting for maternal blood pressure and BMI did not affect the results.

### Metformin vs. placebo

We found no significant difference in z-scores between PCOS-met and PCOS-plac in any of the venous liver flow parameters (Table 3). These remained unchanged after adjusting for maternal blood pressure and BMI.

### PCOS vs. reference population

Fetuses in the total PCOS population (PCOS-tot) had reduced UV and DV flow compared to the reference group. Further, normalized UV, DV, PV and total venous liver blood flow were all reduced compared to the reference population (Table 3).

None of the other flow parameters differed between the metformin and placebo groups, or compared to the reference population (Tables 3 and 4)

## Discussion

In the present study, metformin did not significantly modify the reduced fetal liver blood flow observed in PCOS pregnancies. However, venous liver circulation in fetuses in women with PCOS differed from that in fetuses of healthy low-risk pregnancies. The fetal liver plays a major role in regulation of growth and metabolism. It is a site for synthesis of insulin-like growth factor (IGF-1 and IGF-22) [28]. A direct relationship between venous liver blood flow, IGF synthesis and subsequent peripheral cell proliferation has been reported [29]. Sheep models have demonstrated that reduced venous liver blood flow negatively affects fetal growth [29] and vice versa. IGF-secretion is dependent on the venous perfusion of the fetal liver [29], and IGF I and II levels are in turn positively associated with cord blood levels of insulin and birth weight [30]. Fetal venous liver flow affects fetal growth in low-risk pregnancies [17], with metabolic consequences detectable in childhood [31]. Both accelerated and restricted growth have been studied: macrosomic fetuses of non-diabetic women can rely on seemingly unrestricted umbilical venous supply to the fetal liver [15]. In the other side, venous liver flow is reduced in growth restriction due to reduced placental return, and a relative increase in DV shunting [25, 32].

**Table 3. Fetal venous liver flow parameters in PCOS pregnancies according to metformin/placebo treatment and compared to a low-risk reference population.** Significant values in bold.

| | mean z-score (95%CI) | | | mean z-score (95%CI) | | |
|---|---|---|---|---|---|---|
| | PCOS-met N = 26 | PCOS-plac N = 27 | p-values Met vs plac | PCOS-tot N = 53 | Reference N = 563 | p-values PCOS-tot vs ref |
| UV flow | -0.61 (-1.19–0.00) | -0.30 (-0.73–0.17) | 0.718 | -0.45 (-0.81 - -0.03) | 0.00 (-0.08–0.08) | **0.037** |
| Normalized UV flow | -1.02 (-1.59 - -0.44) | -0.59 (-1.04 - -0.10) | 0.671 | -0.79 (-1.13 - -0.41) | 0.00 (-0.08–0.09) | **<0.001** |
| DV flow | -0.71 (-1.23 - -0.24) | -0.39 (-0.88–0.09) | 0.356 | -0.55 (-0.89 - -0.21) | 0.00 (-0.08–0.09) | **0.003** |
| Normalized DV flow | -0.67 (-0.96 - -0.34) | -0.42 (-0.71 - -0.10) | 0.311 | -0.54 (-0.76 - -0.32) | 0.01 (-0.08–0.09) | **<0.001** |
| PV flow | -0.57 (-1.24–0.16) | -0.16 (-0.62–0.31) | 0.158 | -0.36 (-0.77–0.07) | 0.01 (-0.07–0.10) | 0.099 |
| Normalized PV flow | -1.07 (-1.72 - -0.41) | -0.51 (-1.00–0.01) | 0.117 | -0.78 (-1.19 - -0.35) | 0.01 (-0.07–0.10) | **0.001** |
| Total venous liver flow | -0.32 (-0.90–0.30) | -0.03 (-0.40–0.39) | 0.647 | -0.17 (-0.53–0.19) | -0.01 (-0.10–0.08) | 0.371 |
| Normalized total venous liver | -0.66 (-1.23 - -0.17) | -0.29 (-0.62–0.06) | 0.462 | -0.47 (-0.79 - -0.16) | 0.01 (-0.08–0.10) | **0.006** |
| DV, shunt fraction* | -0.24 (-0.76–0.35) | -0.32 (-0.72–0.07) | 0.369 | -0.28 (-0.62–0.06) | 0.00 (-0.08–0.09) | 0.084 |
| Umbilical venous liver flow | -0.45 (-1.11–0.22) | -0.06 (-0.42–0.40) | 0.422 | -0.25 (-0.61–0.15) | -0.01 (-0.09–0.08) | 0.286 |
| PV fraction of total venous liver flow | -0.20 (-0.91–0.65) | -0.05 (-0.66–0.45) | 0.341 | -0.12 (-0.58–0.35) | 0.00 (-0.08–0.09) | 0.610 |
| Left PV velocity | -0.29 (-0.74–0.20) | 0.09 (-0.38–0.55) | 0.533 | -0.09 (-0.42–0.25) | 0.00 (-0,08–0.09) | 0.803 |

Independent sample t-test

Data are presented as adjusted results for BMI and maternal blood pressure at inclusion.

UV–Umbilical vein

EFW- Estimated fetal weight

DV–Ductus venosus

PV–Portal vein

*(% of umbilical venous blood shunted through DV)

Normalized = Adjusted to estimated fetal weight at time of examination.

## The metformin-effect

Blood flow studies in PCOS pregnancies have been limited to the assessment of the maternal and fetal sides of the placenta (uterine and umbilical artery pulsatility index (PI)). Some indicated a fall in uterine artery pulsatility index from the first to the early second trimester in metformin treated women compared to placebo or no treatment [33, 34], whereas others found no difference [35, 36]. Metformin did not alter the placental resistance in the second half of pregnancy in PCOS women, either on the maternal or the fetal side [33].

Contrary to our hypothesis, the detailed assessment of the fetal venous liver circulation in the present study of PCOS pregnancies found no effect of metformin compared to placebo-exposure.

We have previously reported that metformin exposed fetuses of PCOS mothers, displayed larger head circumferences as new-borns, and maternal BMI modified the metformin effect [37]. The present study was not large enough to perform sub-group analyses according to maternal BMI.

An 8-year follow-up of metformin-exposed children showed increased central fat distribution and obesity, both being cardio-metabolic risk factors [11]. This indicates an epigenetic alteration in utero, affecting the offspring's development. As metformin compared to placebo

apparently did not alter the fetal blood flow, the observed long-term effects of metformin are probably mediated by other mechanisms, or not revealed in the study due to an insufficient sample size.

## The PCOS-effect

The umbilical blood flow from the placenta to the fetus in women with PCOS was impaired compared to a low-risk reference population (Fig 2), although the placental resistance was not increased, i.e. comparable pulsatility index in the uterine artery (Table 4). Surprisingly, the reduced UV return of oxygen and nutrient rich blood observed in PCOS pregnancies was not linked to increased shunting through the DV, as is seen in fetal growth restriction (caused by placental insufficiency) [38]. Further, the reduced UV flow observed in PCOS pregnancies was not accompanied by a reduction in birth weight.

A recent study showed heterogeneity in fetal growth in PCOS, with an increase of both large for gestational age (LGA) and small for gestational age (SGA) new-borns [39]. LGA and SGA subgroups of fetuses might reflect different suboptimal pathways in placental development. Placentas of lean PCOS women without a history of infertility had lower weight, disturbed villous maturation, more fibrosis and decreased mitotic activity compared to controls [40], in addition to higher rates of placenta dysfunction [41]. In contrast, the PCOS population in the present study was unselected and heterogeneous, with an increased prevalence of obesity and gestational diabetes, known to have additional detrimental effects on placental development [42, 43]. Considering fetal weight at examination, total venous liver blood flow was reduced in PCOS pregnancies. Reduced liver blood flow and fetal redistribution are seen in fetal growth restriction [44], however in fetal growth restriction, decreased umbilical perfusion of the liver is accompanied by an increased portal contribution to venous liver flow [32]. In contrast, in PCOS pregnancies, we observed that portal blood flow (from the intestine, spleen and pancreas) was decreased relative to fetal weight. Thus, the fetal circulatory adaptation in PCOS pregnancies seems to differ from that in fetal growth restriction.

The strengths of our study are the prospective, randomized design and high treatment compliance [6]. All ultrasound and Doppler measurements were carried out by one operator (JK), blinded for randomisation. The reproducibility of the measurement techniques has been shown, and identical techniques were applied in the current study, by the same operator who established the reference ranges. Further, there was a high success rate of flow measurements in both study arms decreasing the chance of bias.

The limitation of the study is lack of a power calculation for the blood flow variables. However, we based our sample size on previous research on high-risk conditions, such as fetal growth restriction, that demonstrated significant differences in venous liver blood flow with a similar number of participants [15, 44]. In the light of non-significant difference between

**Table 4. Fetal and maternal flow parameters in PCOS pregnancies according to metformin/placebo treatment.**

|  | PCOS-met N = 26 | PCOS-plac N = 27 | p-values met vs. plac | PCOS total N = 53 |
|---|---|---|---|---|
| MCA PI |  |  |  |  |
| **z-score mean ±SD** | 0.05 ±1.08 | 0.16 ±0.76 | 0.661 | 0.11 ±0.92 |
| Uterine artery PI |  |  |  |  |
| **z-score mean ±SD** | -0.03 ±1.36 | -0.46 ±1.08 | 0.195 | -0.25 ±1.23 |
| Umbilical artery PI |  |  |  |  |
| **z-score mean ±SD** | -0.07 ±1.00 | 0.03 ±0.91 | 0.702 | -0.02 ±0.95 |

Independent sample t-test.

metformin and placebo in our study, but apparently more pronounced flow deviation in the metformin group, we cannot exclude that a potential effect of metformin may be revealed by a larger sample size.

On the other hand, significant results as observed in the comparison between all PCOS pregnancies and the reference population could also be by chance and due to a small sample size (type I error). Further, we are performing multiple analyses, increasing the risk that some results are significant by chance. Since our results were consistent for different flow variables and became more pronounced by normalising flow for fetal weight, we decided to keep the level of significance at $p < 0.05$. The slightly skewed randomization on maternal blood pressure and BMI, in the present sub-study is another weakness. Obesity often accompanies PCOS and since the reference population is a low-risk population, matching the BMI between the two groups (Circmet versus reference) were not possible. Longitudinal measurements of blood flow at several time points in pregnancy would have been an advantage compared to our cross sectionala approach. The reference population was also examined some years earlier when smoking in pregnancy was more common, however smokers were evenly distributed in a scatterplot and not shifting the reference values either way.

## Conclusion

Maternal PCOS-status substantially modified the fetal venous liver circulation in the third trimester, with no further influence of metformin treatment. This may contribute to the adverse long-term metabolic effects observed in children of PCOS mothers.

Our findings should prompt further research on PCOS pregnancies to clarify the role of maternal BMI and GDM-status on the fetal circulation.

## Supporting information

**S1 Checklist. CONSORT checklist.**
(DOC)

**S1 File. Circmet protocol.**
(DOCX)

**S2 File. Pregmet 2 protocol.**
(PDF)

**S1 Data.**
(SAV)

## Acknowledgments

We thank the research participants for their time spent helping to bring new knowledge into the field of medicine.

## Author Contributions

**Conceptualization:** Cathrine Ebbing, Eszter Vanky, Jørg Kessler.

**Data curation:** Sindre Grindheim, Cathrine Ebbing, Henriette Odland Karlsen, Svein Magne Skulstad, Francisco Gómez Real, Marianne Lønnebotn, Tone Løvvik.

**Formal analysis:** Sindre Grindheim, Jørg Kessler.

**Funding acquisition:** Eszter Vanky.

**Investigation:** Svein Magne Skulstad, Tone Løvvik, Eszter Vanky, Jørg Kessler.

**Methodology:** Cathrine Ebbing, Henriette Odland Karlsen, Svein Magne Skulstad, Francisco Gómez Real, Jørg Kessler.

**Project administration:** Tone Løvvik, Eszter Vanky, Jørg Kessler.

**Resources:** Eszter Vanky, Jørg Kessler.

**Supervision:** Eszter Vanky, Jørg Kessler.

**Validation:** Sindre Grindheim, Tone Løvvik, Eszter Vanky, Jørg Kessler.

**Visualization:** Sindre Grindheim.

**Writing – original draft:** Sindre Grindheim, Jørg Kessler.

**Writing – review & editing:** Cathrine Ebbing, Henriette Odland Karlsen, Svein Magne Skulstad, Francisco Gómez Real, Marianne Lønnebotn, Tone Løvvik, Eszter Vanky.

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
