## [Decision Letter · Decision Letter 0]

15 Jun 2021

PONE-D-20-40968

Metformin exposure, maternal PCOS status and fetal venous liver circulation: a randomized, placebo-controlled study

PLOS ONE

Dear Dr. Grindheim,

Thank you for submitting your manuscript to PLOS ONE. After careful consideration, we feel that it has merit but does not fully meet PLOS ONE’s publication criteria as it currently stands. Therefore, we invite you to submit a revised version of the manuscript that addresses the points raised during the review process.

The reviewers have raised a number of issues and specifically on the statistical analysis that require addressing.

We look forward to receiving your revised manuscript.

Kind regards,

Stephen L Atkin, MD

Academic Editor

PLOS ONE

Journal Requirements:

Reviewers' comments:

Reviewer's Responses to Questions

**Comments to the Author**

1. Is the manuscript technically sound, and do the data support the conclusions?

Reviewer #1: Partly

Reviewer #2: Yes

2. Has the statistical analysis been performed appropriately and rigorously? 

Reviewer #1: No

Reviewer #2: Yes

3. Have the authors made all data underlying the findings in their manuscript fully available?

Reviewer #1: Yes

Reviewer #2: Yes

4. Is the manuscript presented in an intelligible fashion and written in standard English?

Reviewer #1: Yes

Reviewer #2: Yes

5. Review Comments to the Author

Reviewer #1: The manuscript entitled ‘Metformin exposure, maternal PCOS status and fetal venous liver circulation: a randomized, placebo-controlled study’ with the aims to compare 75 fetal venous liver blood flow in both metformin versus placebo-exposed fetuses of mothers with PCOS, and all fetuses of PCOS mothers to a low-risk reference population.

This is quite an interesting study, however, the manuscript presentation can be further improved.

Comments

Material and Methods

Line 129 and 161, Table 1 and Table 2 to be placed in the results section.

Statistical analysis

Line 208, proper citation for statistical software including publisher name to be included.

Ensure all statistical tests mentioned in the statistical analysis section are clearly denoted in the results section table(s). 1 or 2-tailed test to be stated.

Results

Line 228-230, for the crude and adjusted analysis, the tables to be clearly indicated. The adjusted variables maternal blood pressure, BMI, and gestational weight gain to be denoted in the table footnote.

Information on whether multicollinearity was assessed between the covariates to be stated.

Line 128 Table 1, for some p values, technically p value cannot be zero (to use symbol p<) Symbol ¶ to be denoted in the table. Since the p value is written on the top of the table, individual ‘p=’ to be omitted. Likewise with Table 3.

Line 161 Table 2, the decimal points for the p value to be standardized.

Nonetheless, based on CONSORT statement, all statistical tests for baseline group comparison to be avoided (Table 1 & Table 2).

Table 1 & 2, at least 1 decimal point for percentage figures.

Line 243 Table 3, what the figures in bracket represents to be clearly denoted. More information on the GLM to be clearly highlighted/denoted e.g dependent variable(s), factors/covariates, multiple comparison correction (if any) etc, Adjusted variables to be indicated in the table footnote and 95%CI to be included for the p values.

Line 253 Table 4, mean and sd to be clearly denoted.

Figure 1 requires improvement by incorporating details of study design i.e. number of groups in the study.

Reviewer #2: Please use the space provided to explain your answers to the questions above. You may also include additional comments for the author, including concerns about dual publication, research ethics, or publication ethics. (Please upload your review as an attachment if it exceeds 20,000 characters) (Limit 200 to 20000 Characters)

please clarify the number of patients enrolled to this particular study in the abstract and put to the abstract that this paper is a part of larger study

6. PLOS authors have the option to publish the peer review history of their article (what does this mean?). If published, this will include your full peer review and any attached files.

Reviewer #1: No

Reviewer #2: **Yes: **Tomasz Milewicz M.D., Ph.D.

---

## [Author Response · Author response to Decision Letter 0]

24 Jul 2021

Dear Stephen L Atkin

Thank you for the opportunity to improve our manuscript. We highly appreciate the meticulous work of the reviewers and have the following comments:

Review Comments to the Author

Reviewer #1: The manuscript entitled ‘Metformin exposure, maternal PCOS status and fetal venous liver circulation: a randomized, placebo-controlled study’ with the aims to compare 75 fetal venous liver blood flow in both metformin versus placebo-exposed fetuses of mothers with PCOS, and all fetuses of PCOS mothers to a low-risk reference population.

This is quite an interesting study, however, the manuscript presentation can be further improved.

Comments

Material and Methods

Line 129 and 161, Table 1 and Table 2 to be placed in the results section.

Response: We chose to place these tables in the M&M section due to the nature of their content, as well as that is the place where they were first referred to. These tables show the baseline of the women included in the current study, as well as the reference group, in addition to maternal and fetal outcomes. Moving this information to the result section could indicate that the reference group was also recruited in parallel to this study, which was not the case. Also, PLOS-one manuscript formatting guidelines state that “Tables should be included directly after the paragraph in which they are first cited.” 

We do not mind moving these tables to the result section, but since the PLOS guidelines statement regarding tables are as such, we ask the editor for advice on where these tables should be placed.

Statistical analysis

Line 208, proper citation for statistical software including publisher name to be included.

Response: The proper citation has been done according to publisher (https://www.ibm.com/support/pages/how-cite-ibm-spss-statistics-or-earlier-versions-spss)

Ensure all statistical tests mentioned in the statistical analysis section are clearly denoted in the results section table(s). 1 or 2-tailed test to be stated.

Response: The 2-tailed p-values have been stated in the M&M and relevant tests have been denoted in all tables.

Results

Line 228-230, for the crude and adjusted analysis, the tables to be clearly indicated. The adjusted variables maternal blood pressure, BMI, and gestational weight gain to be denoted in the table footnote.

Response: We present the crude/unadjusted table since no changes were found after adjustment for these factors. This is now clarified in the table (table 3) and in the manuscript.

We further decided to remove gestational weight gain from the adjustment since this was more likely to be an outcome of metformin treatment and not a baseline parameter. Since BMI at inclusion was border significant only, we ran the adjusted analyzes, however it had no impact on the results.

Information on whether multicollinearity was assessed between the covariates to be stated.

Response: We performed appropriate statistical analyses and found multicollinearity for systolic and diastolic blood pressure. We therefore adjusted for each of them separately as well as combined without finding any influence on the results (overlapping confidence intervals). We therefore chose not to present the adjusted numbers.

Line 128 Table 1, for some p values, technically p value cannot be zero (to use symbol p<) Symbol ¶ to be denoted in the table. Since the p value is written on the top of the table, individual ‘p=’ to be omitted. Likewise with Table 3.

Response: The duplications have been removed (p=) and correct values have been inserted (p<). Symbol ¶ has been denoted in table 1.

Line 161 Table 2, the decimal points for the p value to be standardized.

Response: Three decimals have been standardized in all analyzes

Nonetheless, based on CONSORT statement, all statistical tests for baseline group comparison to be avoided (Table 1 & Table 2).

Response: Since this study was a substudy of a larger RCT, we decided to run these analyses between the groups since the differences found do have a physiological potential to affect the fetal liver blood flow. These differences in baseline were not found in the main study thus indicating a slightly skewed inclusion. We therefore ask not to omit the statistical comparison.

Table 1 & 2, at least 1 decimal point for percentage figures.

Response: Adjustments have been performed accordingly.

Line 243 Table 3, what the figures in bracket represents to be clearly denoted. More information on the GLM to be clearly highlighted/denoted e.g dependent variable(s), factors/covariates, multiple comparison correction (if any) etc, Adjusted variables to be indicated in the table footnote and 95%CI to be included for the p values.

Response: We have clarified the content of the brackets.

We used GLM using each flow measurement (z-score) as a dependent variable, randomization as fixed factor and BMI, systolic and diastolic blood pressure as covariates, first individually, then combined. This is further elaborated in the M&M section.

Line 253 Table 4, mean and sd to be clearly denoted.

Response: The denotation has been revised accordingly.

Figure 1 requires improvement by incorporating details of study design i.e. number of groups in the study.

Response: An updated consort diagram has been incorporated in the manuscript clarifying the randomization groups.

Reviewer #2: Please use the space provided to explain your answers to the questions above. You may also include additional comments for the author, including concerns about dual publication, research ethics, or publication ethics. (Please upload your review as an attachment if it exceeds 20,000 characters) (Limit 200 to 20000 Characters)

please clarify the number of patients enrolled to this particular study in the abstract and put to the abstract that this paper is a part of larger study

Response: This request has been attended to and the text is revised accordingly.

---

## [Decision Letter · Decision Letter 1]

11 Aug 2021

PONE-D-20-40968R1

Metformin exposure, maternal PCOS status and fetal venous liver circulation: a randomized, placebo-controlled study

PLOS ONE

Dear Dr. Grindheim,

Thank you for submitting your manuscript to PLOS ONE. After careful consideration, we feel that it has merit but does not fully meet PLOS ONE’s publication criteria as it currently stands. Therefore, we invite you to submit a revised version of the manuscript that addresses the points raised during the review process.

please address the reviewers final comments

We look forward to receiving your revised manuscript.

Kind regards,

Stephen L Atkin, MD

Academic Editor

PLOS ONE

Journal Requirements:

Reviewers' comments:

Reviewer's Responses to Questions

**Comments to the Author**

1. If the authors have adequately addressed your comments raised in a previous round of review and you feel that this manuscript is now acceptable for publication, you may indicate that here to bypass the “Comments to the Author” section, enter your conflict of interest statement in the “Confidential to Editor” section, and submit your "Accept" recommendation.

Reviewer #1: (No Response)

2. Is the manuscript technically sound, and do the data support the conclusions?

Reviewer #1: Partly

3. Has the statistical analysis been performed appropriately and rigorously? 

Reviewer #1: No

4. Have the authors made all data underlying the findings in their manuscript fully available?

Reviewer #1: Yes

5. Is the manuscript presented in an intelligible fashion and written in standard English?

Reviewer #1: Yes

6. Review Comments to the Author

Reviewer #1: Minor comments

Table 1 Parity & GDM, decimal points for percentage figures.

It is best to present the findings in adjusted form even though it has no impact on the results.

7. PLOS authors have the option to publish the peer review history of their article (what does this mean?). If published, this will include your full peer review and any attached files.

Reviewer #1: No

---

## [Author Response · Author response to Decision Letter 1]

19 Aug 2021

Dear Stephen L Atkin

Thank you for the important feedback we have received, especially from the reviewers, on our last revision. We have attended the last comments and submit now the latest version.

Sincerely

On behalf of the authors

Sindre Grindheim

Review Comments to the Author

Reviewer #1: Minor comments

Table 1 Parity & GDM, decimal points for percentage figures.

Comment: This has been adjusted accordingly.

It is best to present the findings in adjusted form even though it has no impact on the results.

Comment: The adjusted results are now presented in the table and adjusted in the text.

---

## [Editor Report · Decision Letter 2]

11 Jan 2022

Metformin exposure, maternal PCOS status and fetal venous liver circulation: a randomized, placebo-controlled study

PONE-D-20-40968R2

Dear Dr. Grindheim,

We’re pleased to inform you that your manuscript has been judged scientifically suitable for publication and will be formally accepted for publication once it meets all outstanding technical requirements.

Kind regards,

Stephen L Atkin, MD

Academic Editor

PLOS ONE
---

## [Editor Report · Acceptance letter]

20 Jan 2022

PONE-D-20-40968R2 

Metformin exposure, maternal PCOS status and fetal venous liver circulation: a randomized, placebo-controlled study 

Dear Dr. Grindheim:

I'm pleased to inform you that your manuscript has been deemed suitable for publication in PLOS ONE. Congratulations! Your manuscript is now with our production department. 

Kind regards, 

on behalf of

Dr. Stephen L Atkin 

Academic Editor

PLOS ONE